# Fault-tolerant operation of a logical qubit in a diamond quantum processor

M. H. Abobeih[1,2], Y. Wang[1], J. Randall[1,2], S. J. H. Loenen[1,2], C. E. Bradley[1,2], M. Markham[3], D. J. Twitchen[3], B. M. Terhal[1,4] & T. H. Taminiau[1,2 ✉]

Solid-state spin qubits is a promising platform for quantum computation and quantum networks[1,2]. Recent experiments have demonstrated high-quality control over multi-qubit systems[3–8], elementary quantum algorithms[8–11] and non-fault-tolerant error correction[12–14]. Large-scale systems will require using error-corrected logical qubits that are operated fault tolerantly, so that reliable computation becomes possible despite noisy operations[15–18]. Overcoming imperfections in this way remains an important outstanding challenge for quantum science[15,19–27]. Here, we demonstrate fault-tolerant operations on a logical qubit using spin qubits in diamond. Our approach is based on the five-qubit code with a recently discovered flag protocol that enables fault tolerance using a total of seven qubits[28–30]. We encode the logical qubit using a new protocol based on repeated multi-qubit measurements and show that it outperforms non-fault-tolerant encoding schemes. We then fault-tolerantly manipulate the logical qubit through a complete set of single-qubit Clifford gates. Finally, we demonstrate flagged stabilizer measurements with real-time processing of the outcomes. Such measurements are a primitive for fault-tolerant quantum error correction. Although future improvements in fidelity and the number of qubits will be required to suppress logical error rates below the physical error rates, our realization of fault-tolerant protocols on the logical-qubit level is a key step towards quantum information processing based on solid-state spins.

Large-scale quantum computers and quantum networks will require quantum error correction to overcome inevitable imperfections[15–19]. The central idea is to encode each logical qubit of information into several physical data qubits. Non-destructive multi-qubit measurements, called stabilizer measurements, can then be used to identify and correct errors[15–18]. If the error rates of all the components are below a certain threshold, it becomes possible to perform arbitrarily large quantum computations by encoding into increasingly more physical qubits[15,17,18]. A crucial requirement is that all logical building blocks, including the error-syndrome measurement, must be implemented fault tolerantly. At the lowest level, this implies that any single physical error should not cause a logical error.

Over the past several years, steps towards fault-tolerant quantum error correction have been made using spin qubits in silicon[6–8] and in diamond[13,14], as well as in various other hardware platforms, such as superconducting qubits[23–27] and trapped-ion qubits[20,21,31,32]. Pioneering experiments have demonstrated codes that can detect but not correct errors[22,25,26,33], quantum error-correction protocols that can correct only one type of error[13,14,34], as well as non-fault-tolerant quantum error-correction protocols[20,24,34,35]. A recent experiment with trapped-ion qubits has demonstrated the fault-tolerant operation of an error-correction code, albeit through destructive stabilizer measurements and post-processing[21].

In this work, we realize fault-tolerant encoding, gate operations and non-destructive stabilizer measurements for a logical qubit of a quantum error-correction code. Our logical qubit is based on the five-qubit code and we use a total of seven spin qubits in a diamond quantum processor (Fig. 1). Fault tolerance is made possible through the recently discovered paradigm of flag qubits[28–30]. First, we demonstrate a new fault-tolerant encoding protocol based on repeated multi-qubit measurements, which herald the successful preparation of the logical state. Then, we realize the (non-universal) set of transversal single-qubit Clifford gates. Finally, we demonstrate stabilizer measurements on the logical qubit and include a flag qubit to ensure compatibility with fault tolerance. Our stabilizer measurements are non-destructive, the post-measurement state is available in real time and we use feedforward based on the measurement outcomes. Although the logical qubit fidelities do not yet outperform the constituent physical qubits, these results demonstrate the key components of fault-tolerant quantum error correction in a solid-state spin-qubit processor.

## The logical qubit

Stabilizer error-correction codes use auxiliary qubits to perform repeated stabilizer measurements that identify errors. A key requirement for fault tolerance is to prevent errors on the auxiliary qubits from spreading to the data qubits and causing logical errors[18,28] (Fig. 1b). The paradigm of flag fault tolerance provides a solution with minimal qubit overhead[28–30]. Auxiliary qubit errors that would

[1]QuTech, Delft University of Technology, Delft, The Netherlands. [2]Kavli Institute of Nanoscience Delft, Delft University of Technology, Delft, The Netherlands. [3]Element Six, Didcot, UK. [4]JARA Institute for Quantum Information, Forschungszentrum Juelich, Juelich, Germany. ✉e-mail: T.H.Taminiau@TUDelft.nl

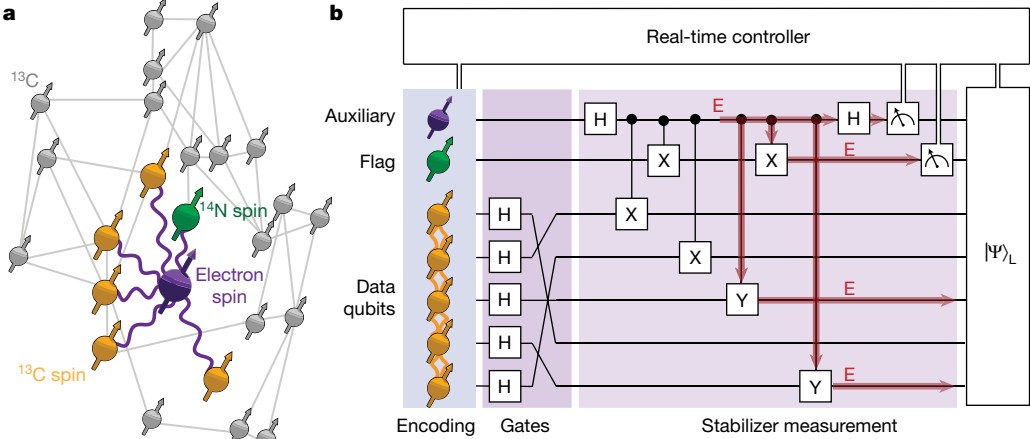

**Fig. 1 | Diamond quantum processor, logical qubit and fault tolerance.**
**a**, Our processor consists of a single NV centre and 27 [13]C nuclear-spin qubits, for which the lattice sites and qubit–qubit interactions are known[38]. We select five [13]C qubits as data qubits that encode the logical state (yellow). The other qubits (grey) are not used here. We use the NV electron spin (purple) as an auxiliary qubit for stabilizer measurements and the NV [14]N nuclear spin (green) as a flag qubit to ensure fault tolerance. Purple lines indicate the electron–nuclear two-qubit gates used here (Methods). Grey lines indicate dipolar

nuclear–nuclear couplings greater than 6 Hz. **b**, Illustration of the main components of the experiment. We realize fault-tolerant encoding, gates and stabilizer measurements with real-time processing on a logical qubit of the five-qubit quantum error-correction code. To ensure that any single fault does not cause a logical error, an extra flag qubit is used to identify errors that would propagate to multi-qubit errors and corrupt the logical state[28]. An illustration of such an error E is shown in red.

cause logical errors are detected using extra flag qubits, so that they can be subsequently corrected (Fig. 1b).

Our logical qubit is based on the five-qubit code, the smallest distance-3 code which can correct any single-qubit error[35,36]. Any logical state is a simultaneous +1 eigenstate of the four stabilizers $s_1 = XXYIY$, $s_2 = YXXYI$, $s_3 = IYXXY$ and $s_4 = YIYXX$, and the logical operators are $X_L = XXXXX$ and $Z_L = ZZZZZ$. Because any error on a single data qubit corresponds to a unique 4-bit syndrome, given as the eigenvalues of the stabilizers, arbitrary single-qubit errors can be identified and corrected. Combined with an auxiliary qubit for stabilizer measurements and a flag qubit to capture harmful auxiliary qubit errors, this makes fault-tolerant error correction possible using seven qubits in total[28].

## System: spin qubits in diamond

Our processor consists of a single nitrogen-vacancy (NV) centre and its surrounding nuclear-spin environment at 4 K (Fig. 1a). These spins are high-quality qubits with coherence times up to seconds for the NV electron spin[37] and minutes for the nuclear spins[3]. The NV electron spin can be read out optically, couples strongly to all other spins and is used as an auxiliary qubit for stabilizer measurements[3,14] (Methods). We use the intrinsic [14]N nuclear spin as the flag qubit. Unlike the other qubits, the flag qubit does not need to maintain coherence during the optical readout. In this device, 27 [13]C nuclear-spin qubits and their lattice positions have been characterized, so that the 406 qubit–qubit interactions are known[38]. Each [13]C qubit can be controlled individually owing to their distinct couplings to the NV electron spin (Methods). Here we use five of the [13]C spin qubits as the data qubits to encode the logical qubit.

A challenge for controlling such a quantum processor is that the spins continuously couple to each other. We realize selective control gates through various echo sequences that isolate interactions between the targeted spins, while also protecting them from environmental decoherence. For all two-qubit gates, we use previously developed electron–nuclear gates, which are based on decoupling sequences on the electron spin[3] (Methods). Furthermore, we introduce interleaved and asynchronous echo stages that cancel unwanted couplings between the data qubits (Methods). These extra echo stages are essential for the relatively long gate sequences realized here.

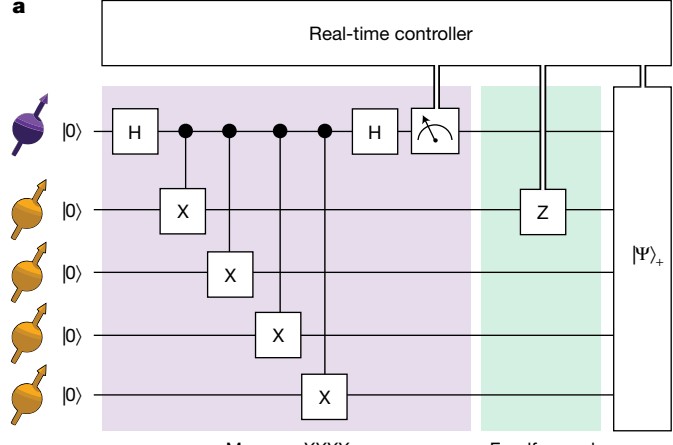

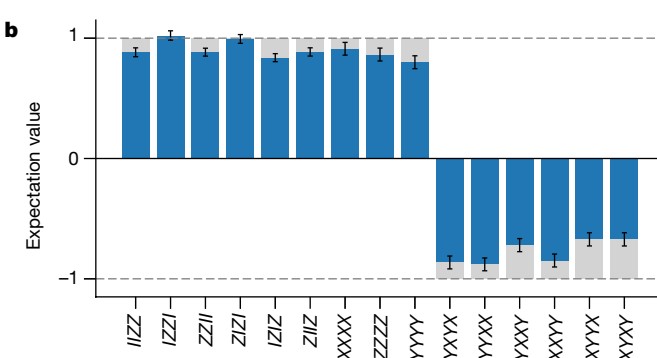

**Fig. 2 | Non-destructive stabilizer measurements with real-time feedforward. a**, Circuit diagram for the deterministic preparation of a four-qubit GHZ entangled state ($|\psi\rangle_+ = (|0000\rangle + |1111\rangle)/\sqrt{2}$) using a measurement of the stabilizer *XXXX*. **b**, Measured expectation values of the 15 operators that define the ideal state. The obtained fidelity with the target state is 0.86(1), confirming genuine multipartite entanglement. Grey bars show the ideal expectation values. Error bars are one standard deviation.

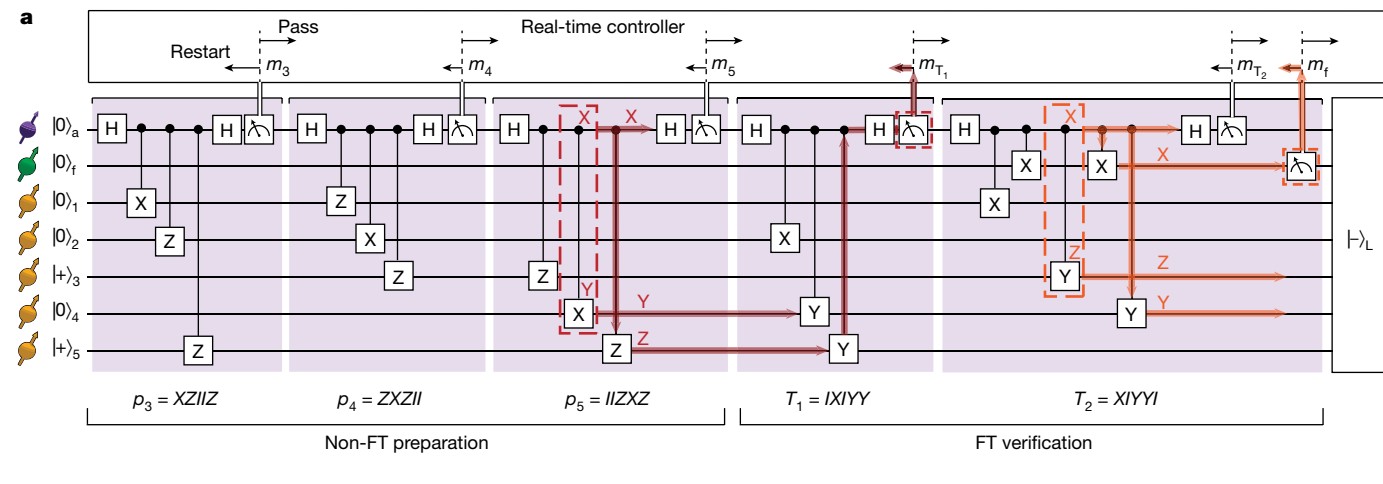

**a**

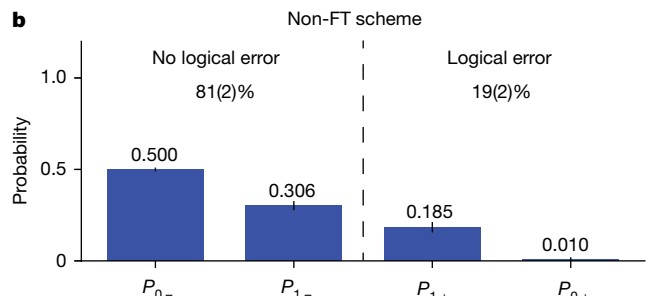

**b**

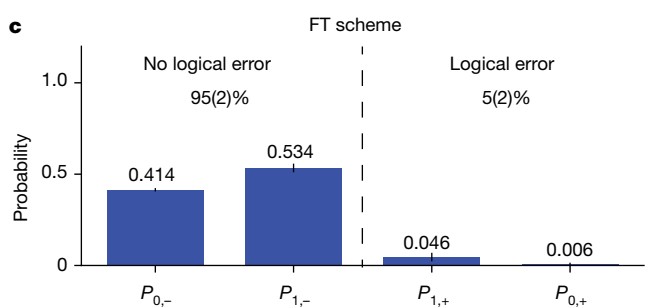

**c**

**Fig. 3 | Fault-tolerant encoding of the logical qubit. a**, Encoding circuit. The first stage prepares $|-\rangle_L$ non-fault-tolerantly ('non-FT preparation') by starting with $|00+0+\rangle$ (an eigenstate of $p_1, p_2$) and measuring the logical operators $p_3$ to $p_5$. The second 'FT verification' stage consists of two stabilizer measurements, $T_1 = p_2 \cdot p_4 \cdot p_5$, $T_2 = p_1 \cdot p_3 \cdot p_5$, and a flag qubit measurement. Echo sequences are inserted between the measurements to decouple the qubits (not shown, see Supplementary Figs. 8 and 9). Successful preparation is heralded by satisfying a set of conditions for the measurement outcomes (see main text). Red indicates an example of an auxiliary qubit fault (an XY error in a two-qubit gate)

that would propagate to a logical error but is detected by the $T_1$ verification step. Orange indicates an example of a single fault in the verification stage that would propagate into a logical error but is detected by the flag qubit. **b,c**, Probabilities to obtain the desired logical state $|-\rangle_L$ without error ($P_{0,-}$) or with a single-qubit Pauli error ($P_{1,-}$), and the probabilities to obtain the opposite logical state $|+\rangle_L$ with zero error ($P_{0,+}$) or with a single-qubit Pauli error ($P_{1,+}$). Note that $P_{1,\pm}$ are summed over all 15 possible errors. These 32 states are orthogonal and span the full five-qubit Hilbert space.

## Non-destructive stabilizer measurements

We start by demonstrating non-destructive four-qubit stabilizer measurements with real-time feedforward operations based on the measurement outcomes (Fig. 2). Despite the central role of such measurements in many error-correction codes, including the five-qubit code, the Steane code and the surface code[15–18], experimental implementations with feedforward have remained an outstanding challenge.

We benchmark the measurement by using it to deterministically create a four-qubit entangled state. We prepare the state $|0000\rangle$ and measure the operator $XXXX$. This projects the qubits into the Greenberger–Horne–Zeilinger (GHZ) state ($|\psi\rangle_\pm = (|0000\rangle \pm |1111\rangle)/\sqrt{2}$), with the sign determined by the measurement outcome. We process the measurement outcomes in real time using a microprocessor and apply the required correction to deterministically output the state ($|\psi\rangle_+$) with a fidelity of 0.86(1). Because this result is obtained without any post-selection, it highlights that the post-measurement state is available for all measurement outcomes, satisfying one of the key requirements for error correction.

## Fault-tolerant encoding

To prepare the logical qubit, we introduce a new scheme that uses repeated stabilizer measurements and a flag qubit to herald successful preparation (Fig. 3a). In contrast to the scheme introduced by Chao

and Reichardt[28], no direct two-qubit gates between the data qubits are required (fifth section of the Supplementary Information). We demonstrate the preparation of the logical state ($|-\rangle_L = \frac{1}{\sqrt{2}}(|0\rangle_L - |1\rangle_L)$). This state is the unique +1 eigenstate of five independent weight-3 logical-$X$ operators, namely, $p_1 = IZXZI$, $p_2 = ZIIZX$, $p_3 = XZIIZ$, $p_4 = ZXZII$ and $p_5 = IIZXZ$. Therefore, one can prepare $|-\rangle_L$ by initializing the data qubits into the product state $|00+0+\rangle$, which is an eigenstate of $p_1$ and $p_2$, and subsequently measuring $p_3$ to $p_5$ (Fig. 3a). This preparation scheme is not fault tolerant because faults involving the auxiliary qubit can cause weight-2 errors, which can result in logical errors (Fig. 3a). We refer to these steps as the non-FT encoding scheme.

We make the preparation circuit fault tolerant by adding two stabilizer measurements, $T_1 = p_2 \cdot p_4 \cdot p_5 = IXIYY$ and $T_2 = p_1 \cdot p_3 \cdot p_5 = XIYYI$ with a flag qubit (Fig. 3a). Successful preparation is heralded by the following conditions: (1) the measurement outcomes of $T_1$ and $T_2$ are compatible with the measurement outcomes $m_i$ of the logical operators $p_i$, that is, $m_{T_1} = m_2 \times m_4 \times m_5$ and $m_{T_2} = m_1 \times m_3 \times m_5$; (2) the flag is not raised (that is, the flag qubit is measured to be in $|0\rangle$). Otherwise, the state is rejected. The order of two-qubit gates is carefully chosen to ensure fault tolerance while minimizing the number of operations. Further details and a proof of the fault tolerance of this scheme are given in the sixth section of the Supplementary Information. We refer to this preparation as the FT encoding scheme.

To reduce the impact of auxiliary qubit measurement errors[10,14], we also require all stabilizer measurement outcomes to be +1 (that is, the

NV electron spin is measured to be in $|0\rangle$). These outcomes are more reliable[3] (Methods), increasing the fidelity of the state preparation, at the cost of a lower success probability (Supplementary Table 1).

We compare the non-FT and FT encoding schemes. We define the logical state fidelity $F_L$ as (Methods)

$$F_L = \sum_{E \in \varepsilon} \mathrm{Tr}(E|-\rangle_L \langle -|_L E\rho),\qquad(1)$$

in which $\rho$ is the prepared state and $\varepsilon = \{I, X_i, Y_i, Z_i, i = 1, 2, \ldots, 5\}$ is the set of all single-qubit Pauli errors. The fidelity $F_L$ gives the probability that there is at most a single-qubit error in the prepared state, that is, there is no logical error. We characterize the prepared state by measuring the 31 operators that define the target state (Extended Data Fig. 2 and Methods). We find that the FT encoding scheme ($F_L = 95(2)\%$) outperforms the non-FT scheme ($F_L = 81(2)\%$).

To understand this improvement, we analyse the underlying error probability distributions (Figs. 3b,c). For the five-qubit code, the $|-\rangle_L$ state plus any number of Pauli errors is equivalent to either $|-\rangle_L$ with at most one Pauli error (no logical error) or to $|+\rangle_L$ with at most one Pauli error (a logical error). We calculate the overlaps between the prepared state and those states. The results show that the FT scheme suppresses logical errors, consistent with fault tolerance preventing single faults propagating to multi-qubit errors. The overall logical state fidelity $F_L$ is improved, despite the higher probability of single-qubit errors owing to the increased complexity of the sequence.

## Fault-tolerant logical gates

The five-qubit code supports a complete set of transversal single-qubit Clifford gates, which are naturally fault tolerant[16,39]. We apply four transversal logical gates to $|-\rangle_L$ (Fig. 4): $X_L = X_1 X_2 X_3 X_4 X_5$, $Y_L = Y_1 Y_2 Y_3 Y_4 Y_5$, the Hadamard gate $H_L = P_\pi H_1 H_2 H_3 H_4 H_5$ and the phase gate $S_L = P_\pi S_1 S_2 S_3 S_4 S_5$, in which $P_\pi$ is a permutation of the data qubits[16,39] (Fig. 4b). These permutations are fault tolerant because we realize them by relabelling the qubits rather than by using SWAP gates[39]. For completeness, we note that universal computation requires further non-transversal gates, constructed—for example—with auxiliary logical qubits, which are not pursued here[18].

Our control system performs the underlying single-qubit gates by tracking basis rotations and compiling them with subsequent gates or measurements (Methods). In the sequence considered here (Fig. 4a), such compilation does not increase the physical operation count and there is no reduction of fidelity (Fig. 4c). For comparison, we also implement the 'worst-case' scenario, in which the logical gates are applied physically (Fig. 4c). This includes five single-qubit gates and the corresponding extra echo sequences between the state preparation and the measurement stage. Together, the demonstrated transversal logical gates enable the fault-tolerant preparation of all six eigenstates of the logical Pauli operators.

## Fault-tolerant stabilizer measurements

Finally, we demonstrate and characterize a flagged stabilizer measurement on the encoded state (Fig. 5a). Such measurements are a primitive for fault-tolerant quantum error-correction protocols[28]. To ensure that the measurement is compatible with fault tolerance, the two-qubit gates are carefully ordered and a flag qubit is added to capture the auxiliary qubit errors that can propagate to logical errors[28].

We prepare the logical state $|-\rangle_L$ and measure the stabilizer $s_1 = XXYIY$ (Fig. 5a). The resulting output consists of the post-measurement state and two classical bits of information from the measurements of the auxiliary and flag qubits (Fig. 5b). The logical state fidelity $F_L$ is given by the probability that the logical information can be correctly extracted (no logical error) when taking into account the flag measurement

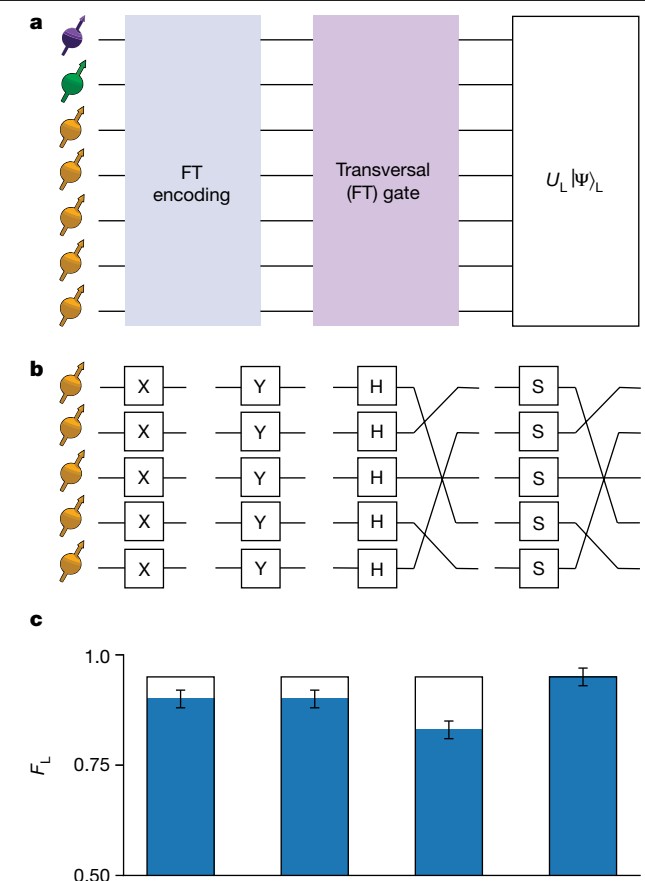

**Fig. 4 | Fault-tolerant gates on the logically encoded qubit. a**, We apply transversal logical gates on the encoded state $|-\rangle_L$ and measure the resulting logical state fidelity $F_L$ (equation (1)) with respect to the targeted state. **b**, Logical $X_L$, $Y_L$, $H_L$ (Hadamard) and $S_L$ ($\pi/2$ rotation around the $z$-axis) are realized by five single-qubit gates. For $H_L$ and $S_L$, this is followed by a permutation of the qubits by relabelling them. **c**, Grey bars indicate logical state fidelity when compiling the logical gates with subsequent operations (0.95(2) for all gates). Blue bars are logical state fidelities after physically applying the transversal logical gates (0.90(2), 0.90(2), 0.83(2), 0.95(2) for $X_L$, $Y_L$, $H_L$ and $S_L$, respectively). Error bars are one standard deviation.

outcome. The interpretation of the error syndrome changes if the flag is raised (Methods). We find $F_L = 0.77(4)$ for the post-measurement state without any post-selection. Higher logical state fidelities can be obtained by post-selecting on favourable outcomes, but this is incompatible with error correction.

To illustrate the benefit of the flag qubit, we compare the logical state fidelities with and without taking the flag measurement outcome into account. Because auxiliary qubit errors that propagate to logical errors are naturally rare, no marked difference is observed (Fig. 5c). Therefore, we introduce a Pauli $Y$ error on the auxiliary qubit (Fig. 5a). This error propagates to the two-qubit error $Y_3 Y_5$. For the case without flag information, this error causes a logical flip $Z_L$ (Methods) and the logical state fidelity drops below 0.5. By contrast, with the flag qubit, this non-trivial error is detected (Fig. 5b) and remains correctable, so that the logical state fidelity is partly recovered (Fig. 5c).

## Conclusion

In conclusion, we have demonstrated encoding, gates and non-destructive stabilizer measurements for a logical qubit of an error-correction code in a fault-tolerant way. Our results advance

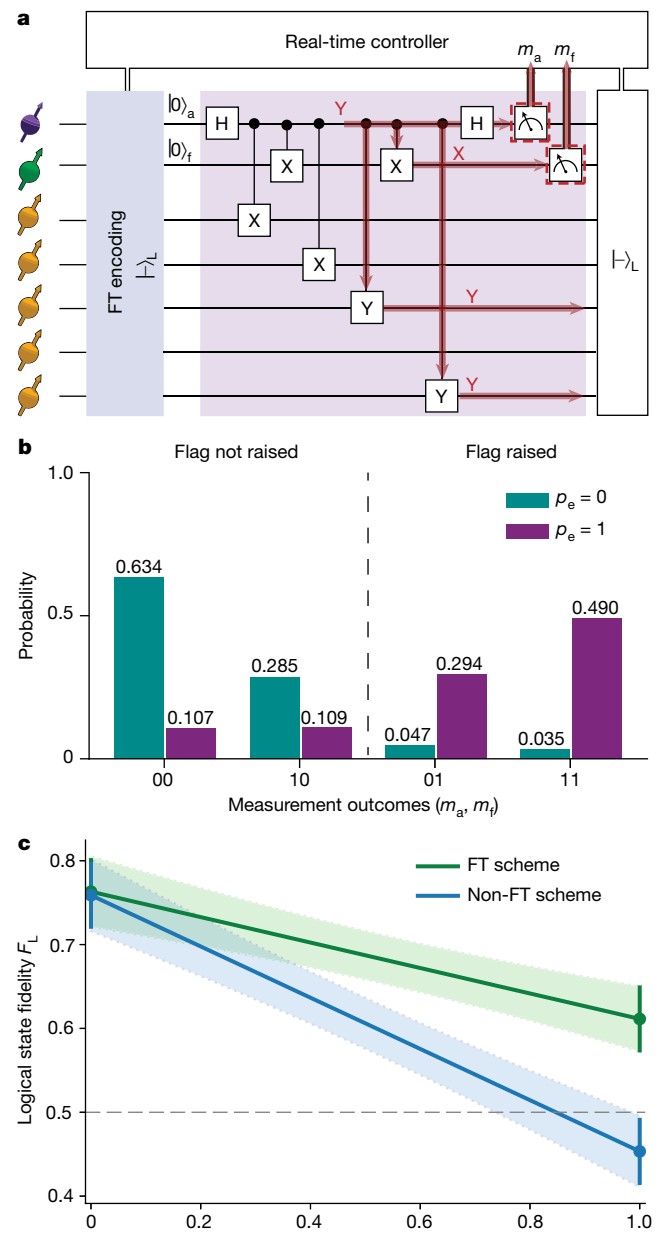

**Fig. 5 | Fault-tolerant stabilizer measurement. a**, Circuit diagram to measure the stabilizer *XXYIY* on the encoded state. As an example to illustrate the compatibility with fault tolerance, we insert a *Y* error on the auxiliary qubit. This error will propagate to the two-qubit error $Y_3Y_5$ on the data qubits, which leads to a logical *Z* error. However, because the error also triggers the flag qubit, it can be accounted for (Methods). **b**, Probability of the measurement outcomes of the auxiliary ($m_a$) and flag ($m_f$) qubits when inserting ($p_e = 1$) or not inserting ($p_e = 0$) the *Y* error on the auxiliary qubit. The results show that the flag qubit successfully detects this error. **c**, Logical state fidelity $F_L$ after the stabilizer measurement as a function of the error probability $p_e$. The non-FT case does not take the flag outcome into account. Values between $p_e = 0$ and $p_e = 1$ are calculated as weighted sums (Methods).

solid-state spin qubits from the physical-qubit level to the logical-qubit level, at which fault-tolerant operations become possible. Such fault tolerance is a necessity for large-scale quantum computation, in which error rates must ultimately be suppressed to extremely low levels.

Future challenges are to perform complete quantum error-correction cycles, encode several logical qubits, realize universal fault-tolerant gates and—ultimately—suppress logical error rates exponentially below

physical error rates. Although the demonstrated operations are of high fidelity—the experiments consist of up to 40 two-qubit gates and eight mid-circuit auxiliary qubit readouts (Fig. 5a)—improvements in both the fidelities and the number of qubits will be required.

Improved gates might be realized through tailored optimal control schemes that leverage the precise knowledge of the system and its environment[40] (Fig. 1a). Coupling to optical cavities can further improve readout fidelities[1,41]. Scaling to large code distances and several logical qubits can be realized through already-demonstrated magnetic[40] and optical[42] NV–NV connections that enable modular, distributed, quantum computation based on the surface code and other error-correction codes[19]. Therefore, our demonstration of the building blocks of fault-tolerant quantum error correction is a key step towards quantum information processing based on solid-state spin qubits.

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

## Methods

### Sample

We use a naturally occurring NV centre in a homo-epitaxially chemical-vapour-deposition-grown diamond with a 1.1% natural abundance of $^{13}$C and a $\langle 111 \rangle$ crystal orientation (grown by Element Six). A solid-immersion lens is used to enhance the photon-collection efficiency[43]. The NV centre has been selected for the absence of $^{13}$C spins with hyperfine couplings >500 kHz. These experiments are performed at a temperature of 4 K, at which the electron-spin relaxation is negligible ($T_1 = 3.6(3) \times 10^3$ s)[37].

### Qubits and coherence times

The NV electron-spin auxiliary qubit is defined between the states $m_s = 0 (|0\rangle)$ and $m_s = -1 (|1\rangle)$. The NV electron-spin coherence times are $T_2^* = 4.9(2)$ μs, $T_2 = 1.182(5)$ ms and up to seconds under dynamical decoupling[37]. The $^{14}$N nuclear-spin flag qubit is defined between the states $m_I = 0 (|0\rangle)$ and $m_I = -1 (|1\rangle)$. The $^{13}$C nuclear-spin data qubits in this device have been characterized in detail in previous work[3,38,44] (Fig. 1a). See Supplementary Tables 2–5 for the hyperfine parameters, coherence times and qubit–qubit interactions for the qubits used here.

### Magnetic field

A magnetic field of about 403 G is applied using a room-temperature permanent magnet on a XYZ translation stage. This applied field lifts the degeneracy of the $m_s = \pm 1$ states owing to the Zeeman term (first section of the Supplementary Information). We stabilize the magnetic field to <3 mG using temperature stabilization and an automatic recalibration procedure (every few hours). We align the magnetic field along the NV axis using thermal echo sequences with an uncertainty of 0.07° in the alignment[38].

### Single-qubit and two-qubit gates

Single-qubit gates and echo pulses are applied using microwave pulses for the NV electron spin ($m_s = 0 \leftrightarrow m_s = -1$ transition, Hermite pulse shapes[37,45], Rabi frequency of about 15 MHz) and using radio-frequency (RF) pulses for the $^{13}$C spin qubits (error function pulse shapes[3], typical Rabi frequency of about 500 Hz) and the $^{14}$N spin qubit (error function pulse shapes, Rabi frequency of about 2 kHz).

The Hermite pulse envelopes of the microwave pulses are defined as

$$A\left[1 - c\left(\frac{t-\mu}{T}\right)^2\right] \cdot \exp\left[-\left(\frac{t-\mu}{T}\right)^2\right], \qquad (2)$$

in which $c = 0.956$ for π pulses and $c = 0.667$ for π/2 pulses, $\mu = 0.5 t_{\text{pulse}}$, $T = 0.1667 t_{\text{pulse}}$, $t_{\text{pulse}}$ is the microwave pulse length and $A$ is the pulse amplitude, which is experimentally calibrated to achieve a π or π/2 rotation. For this work, we use $t_{\text{pulse}} = 168$ ns for π pulses and $t_{\text{pulse}} = 100$ ns for π/2 pulses. The envelope of the RF pulses is defined as

$$f(t) = 1 - \frac{1}{2}\text{erf}\left(\frac{2(\Delta t - t + t_0)}{\Delta t}\right) - \frac{1}{2}\text{erf}\left(\frac{2(\Delta t + t - t_{\text{pulse}})}{\Delta t}\right), \qquad (3)$$

in which $\Delta t$ is the rise time, $t_0$ is the start time of the pulse, $t_{\text{pulse}}$ is the pulse length and $\text{erf}(x)$ is the error function[3]. We ensure that the RF pulses consisted of an integer number of periods of the RF waveform, that is, we ensure that $\omega t_{\text{pulse}} = 2\pi n$ for integer $n$. This ensures that any phase picked up on the electron spin owing to the RF pulse is cancelled. Note that the $^{13}$C spin qubits (data qubits) are distinguishable in frequency owing to their hyperfine coupling to the NV electron spin (Supplementary Table 2).

Electron–nuclear two-qubit gates are realized using two different gate designs, depending on the properties of the targeted nuclear spin. For data qubits 1, 2, 4 and 5, two-qubit gates are realized through dynamical decoupling sequences of $N$ equally spaced π-pulses on the electron spin of the form[3,46] $(\tau_r - \pi - \tau_r)^N$. This design requires a notable hyperfine component perpendicular to the applied magnetic field[46]. For data qubit 3 and the flag qubit (the $^{14}$N spin), the perpendicular hyperfine coupling is small and we perform two-qubit gates by interleaving the dynamical decoupling sequence with RF pulses[3]. Both gate designs simultaneously decouple the NV electron spin from the other qubits and the environment[3]. The parameters and fidelities for the two-qubit gates are given in Supplementary Table 4. Note that direct nuclear–nuclear two-qubit gates can also be constructed[47], but because the natural interaction is much weaker than the electron–nuclear interaction, we don't use such gates here and designed the FT encoding circuit based on electron–nuclear gates only.

### Compilation of gate sequences

Our native two-qubit gates are electron-controlled nuclear-spin rotations and are equivalent to the CNOT gate up to single-qubit rotations (Supplementary Fig. 4). To implement the sequences shown in the figures, we first translate all gates into these native gates and compile the resulting sequence. Afterwards, the circuit is translated into the actual pulse sequence. At the core of this compilation process is the tracking and synchronization of the qubit phases and the corresponding pulse timings. See Supplementary Information for the details of this compilation process (Supplementary Figs. 4–9 and pseudocode 1–8).

### Echo sequences for the data qubits

To mitigate decoherence of the data qubits owing to their spin environment, we use echo sequences that are interleaved throughout the experiments. These echo sequences ensure that the data qubits rephase each time they are operated on. Furthermore, the sequence design minimizes the time that the auxiliary electron-spin qubit is idling in superposition states, which are prone to dephasing. We use two echo stages between stabilizer measurements, as well as before and after the logical gates of Fig. 4, which provides a general and scalable solution for the timing of all gates and echoes (third section of the Supplementary Information).

An extra challenge is that, owing to the length of the sequences (up to 100 ms), we need to account for the small unwanted interactions between the nuclear-spin data qubits. The measured coupling strengths show that the strongest couplings are between qubits 3 and 2 (16.90(4) Hz) and between qubits 3 and 5 (12.96(4) Hz)[38] (Supplementary Table 5). Such interactions can introduce correlated two-qubit errors that are not correctable in the distance-3 code considered here, which can only handle single-qubit errors in the code block.

To mitigate these qubit–qubit couplings, we decouple qubit 3 asynchronously from the other qubits (Supplementary Fig. 8). Ultimately, such local correlated errors can be suppressed entirely by larger distance codes.

### Real-time control and feedforward operations

Real-time control and feedforward operations are implemented through a programmable microprocessor (Jaeger ADwin Pro II) operating on microsecond timescales. The microprocessor detects photon events coming from the detectors, infers the measurement outcomes and controls both the subsequent sequences in the arbitrary waveform generator (Tektronix AWG 5014c) and the lasers for the auxiliary qubit readout. The precise timing for quantum gates (1-ns precision) is based on the clock of the arbitrary waveform generator. Furthermore, the microprocessor operates various control loops that prepare the NV centre in the negative charge state, on resonance with the lasers and in the focus of the laser beam (see second and third sections of the Supplementary Information).

### Readout of the auxiliary qubit

The electron spin (auxiliary qubit) is read out by resonantly exciting the $m_s = 0$ to $E_x$ optical transition[43]. For one or more photons detected, we assign the $m_s = 0$ outcome; for zero photons, we assign $m_s = \pm 1$. The

single-shot readout fidelities are $F_0 = 90.5(2)\%$ and $F_1 = 98.6(2)\%$ for $m_s = 0$ and $m_s = -1$, respectively (average fidelity 94.6(1)%).

Uncontrolled electron-spin flips in the excited state cause dephasing of the nuclear spins through the hyperfine interaction. To minimize such spin flips, we avoid unnecessary excitations by using weak laser pulses, so that a feedback signal can be used to rapidly turn off the laser on detection of a photon (within 2 μs). The resulting probability that the electron spin is in state $m_s = 0$ after correctly assigning $m_s = 0$ in the measurement is 0.992 (ref. [14]).

For measurements that are used for heralded state preparation, that is, for which we only continue on a $m_s = 0$ outcome (see, for example, Fig. 3), we use shorter readout pulses. This improves the probability that a $m_s = 0$ outcome correctly heralds the $m_s = 0$ state, at the cost of reduced success probability (Supplementary Table 1).

## System preparation and qubit initialization
At the start of the experiments, we first prepare the NV centre in its negative charge state and on resonance with the lasers. We then initialize the NV electron spin in the $m_s = 0$ state through a spin pumping process (fidelity > 99.7%)[43]. We define the electron-spin qubit between the states $m_s = 0$ ($|0\rangle$) and $m_s = -1$ ($|1\rangle$). We initialize the data qubits through SWAP sequences (Supplementary Fig. 6) into $|0\rangle$ and subsequent optical reset of the auxiliary qubit (initialization fidelities 96.5–98.5%; see Supplementary Table 4). The flag qubit is initialized through a projective measurement that heralds preparation in $|0\rangle$ (initialization fidelity 99.7%). Other product states are prepared by subsequent single-qubit gates.

## Final readout of the data qubits
Measuring single-qubit and multi-qubit operators of the data qubits is performed by mapping the required correlation to the auxiliary qubit (through controlled rotations) and then reading out the auxiliary qubit[14]. To provide best estimates for the measurements, we correct the measured expectation values (Fig. 2 and Extended Data Figs. 1 and 2) for infidelities in the readout sequence; see Bradley et al.[3] for the correction procedure.

## Fidelity of the GHZ state
The fidelity of the prepared state $\rho$ (in Fig. 2 and Extended Data Fig. 1) with respect to the target GHZ state $(|\psi\rangle_+)$ is obtained as

$$
\begin{aligned}
F = \mathrm{Tr}&(|\psi\rangle_+\langle\psi|_+\rho) \\
= \frac{1}{16}&(1 + \langle IIZZ\rangle + \langle IZZI\rangle + \langle ZZII\rangle + \langle ZIZI\rangle + \langle IZIZ\rangle \\
&+ \langle ZIIZ\rangle + \langle XXXX\rangle + \langle ZZZZ\rangle + \langle YYYY\rangle - \langle YXYX\rangle \\
&- \langle YYXX\rangle - \langle YXXY\rangle - \langle XXYY\rangle - \langle XYYX\rangle - \langle XYXY\rangle).
\end{aligned}
\tag{4}
$$

## Assessing the logical state fidelity
The logical state fidelity $F_L$ is defined in equation (1) and gives the probability that the state is free of logical errors. Said differently, $F_L$ is the fidelity with respect to the ideal five-qubit state after a round of perfect error correction, or the probability to obtain the correct outcome in a perfect fault-tolerant logical measurement. Although fault-tolerant circuits for logical measurement exist[28], we do not experimentally implement these here. Instead, we extract $F_L$ from a set of measurements, as described in the following using $|-\rangle_L$ as an example.

The logical state $|-\rangle_L$ is the unique simultaneous eigenstate of the five weight-3 operators $p_i$ with eigenvalue +1. Thus, we can describe the state $E|-\rangle_L$ (with $E$ a Pauli error) as the projector

$$
E|-\rangle_L\langle-|_L E = \prod_{i=1}^{5} \frac{(1 + m_i p_i)}{2},
$$

in which $m_i = \pm 1$ is the measurement outcome of $p_i$ and $m_i = -1$ when $E$ anticommutes with $p_i$. This projector can be expanded as a summation

of 31 multi-qubit Pauli operators (including a constant), which are listed in Extended Data Fig. 2. The logical state fidelity $p_i$ in equation (1) can then be written as

$$
\begin{aligned}
F_L = &\sum_{E\in\varepsilon} \mathrm{Tr}(E|-\rangle_L\langle-|_L E\rho) \\
= &\frac{1}{2} + \frac{1}{8}(\langle IZXZI\rangle + \langle ZIIZX\rangle + \langle XZIIZ\rangle + \langle ZXZII\rangle + \langle IIZXZ\rangle \\
&+ \langle YIXIY\rangle + \langle IYYIX\rangle + \langle XIYYI\rangle + \langle IXIYY\rangle + \langle YYIXI\rangle + \langle ZZYXY\rangle \\
&+ \langle YXYZZ\rangle + \langle ZYXYZ\rangle + \langle XYZZY\rangle + \langle YZZYX\rangle + \langle XXXXX\rangle).
\end{aligned}
\tag{5}
$$

Here $\varepsilon = \{I, X_i, Y_i, Z_i, i = 1,2, …, 5\}$ is the set of correctable errors for the five-qubit code. To obtain $F_L$ experimentally, we measure this set of expectation values.

## Logical state fidelity with flag
If the flag in the circuit in Fig. 5a is not raised, then a cycle of error correction would correct any single-qubit error on a logical state. The logical state fidelity is then given by equation (1), which we now refer to as $F_L^{\text{not raised}}$. A raised flag leads to a different interpretation of the error syndrome[28] (Supplementary Table 7).

For example, the $Y$ error on the auxiliary qubit in Fig. 5a leads to the output state $Y_3 Y_5|-\rangle_L$, for which the eigenvalues of $s_1 = XXYIY$, $s_2 = YXXYI$, $s_3 = IYXXY$ and $s_4 = YIYXX$ give the syndrome $[+1, -1, -1, -1]$. Without flag, the corresponding single-qubit recovery is $Z_4$, which changes the syndrome back to all +1 (Supplementary Table 7). This recovery leads to the remaining error $Y_3 Z_4 Y_5$, which is a logical $Z$ error. However, taking the flag measurement outcome into account, the syndrome is interpreted differently and the recovery is $Y_3 Y_5$, so that no error is left (Supplementary Table 7).

For the cases in which the flag is raised, the logical state fidelity with respect to $|-\rangle_L$ is now given by:

$$
\begin{aligned}
F_L^{\text{raised}} = &\sum_{E\in\varepsilon'} \mathrm{Tr}(E|-\rangle_L\langle-|_L E\rho) \\
= &\frac{1}{2} + \frac{1}{32}(6\langle IIZXZ\rangle + 6\langle ZXZII\rangle + 6\langle YYIXI\rangle - 2\langle ZIIZX\rangle \\
&+ 6\langle IXIYY\rangle + 2\langle YZZYX\rangle + 2\langle XYZZY\rangle - 2\langle IZXZI\rangle \\
&+ 2\langle ZYXYZ\rangle + 6\langle XIYYI\rangle + 2\langle YXYZZ\rangle + 2\langle ZZYXY\rangle \\
&+ 6\langle IYYIX\rangle - 2\langle YIXIY\rangle + 2\langle XXXXX\rangle - 2\langle XZIIZ\rangle),
\end{aligned}
\tag{6}
$$

with $\varepsilon'$ being another set of correctable errors

$$
\varepsilon' = \{I, X_1, X_3 Y_5, Z_1, X_2, Y_2, Z_3 Y_5, X_1 Y_2, Y_3, Z_3, X_4, Y_4, Y_3 Y_5, X_5, Y_5, X_1 Z_2\}. \tag{7}
$$

A detailed derivation for this set of errors and their corresponding syndromes are given in the fifth section of the Supplementary Information.

The logical state fidelity after the stabilizer measurement (Fig. 5) is calculated as the weighted sum of the fidelities conditioned on the two flag outcomes:

$$
F_L = p_f \cdot F_L^{\text{raised}} + (1 - p_f) \cdot F_L^{\text{not raised}}, \tag{8}
$$

with $p_f$ being the probability that the flag is raised and $F_L^{\text{raised}}$ and $F_L^{\text{not raised}}$ are as defined above.

Finally, to construct the logical state fidelity as a function of $p_e$ (Fig. 5c), we measure $F_L$ with ($p_e = 0$) and without ($p_e = 1$) the auxiliary qubit error and calculate the outcomes for other error probabilities $p_e$ from their weighted sum:

$$
F_L(p_e) = (1 - p_e) \cdot F_L(p_e = 0) + p_e \cdot F_L(p_e = 1) \tag{9}
$$

## Error distribution in the prepared state
The overlaps between the prepared state $\rho$ and the state $E|-\rangle_L$ with $E$ identity or a single-qubit error are written as $P_{0,-}$ and $P_{1,-}$, respectively.

These correspond to the cases that there is no logical error. The overlaps between the prepared state $\rho$ and the state $E|+\rangle_L$ with $E$ identity or a single-qubit error are written as $P_{0,+}$ and $P_{1,+}$, respectively. In these cases, there is a logical error. These overlaps are shown in Fig. 3b,c and calculated as ($\alpha = \pm$)

$$P_{0,\alpha} = \mathrm{Tr}(|\alpha\rangle_L\langle\alpha|_L\rho), \qquad (10)$$

$$P_{1,\alpha} = \sum_{E\in\varepsilon} \mathrm{Tr}(E\,|\alpha\rangle_L\langle\alpha|_L E\rho). \qquad (11)$$

These overlaps can be explicitly expressed in terms of the 31 measured expectation values (see seventh section of the Supplementary Information).

## Error analysis

The uncertainties in the measured fidelities, logical state fidelities and probabilities ($P_{0/1,\pm}$) are obtained from the uncertainties in the measured expectation values using error propagation. For example, the logical state fidelity $F_L$ is calculated as

$$F_L = \frac{1}{2} + \frac{1}{8}\left(\sum_i A_i\right), \qquad (12)$$

in which $A_i$ are the 16 expectation values shown in equation (5). Assuming that the errors in the measured expectation values are independent, the standard deviation in $F_L$ is:

$$\sigma_{F_L} = \frac{1}{8}\left(\sum_i \sigma_{A_i}^2\right)^{\frac{1}{2}}, \qquad (13)$$

in which $\sigma_{A_i}$ is the standard deviation of the expectation value $A_i$ and is given by a binomial distribution[42]. Note that $\sigma_{A_i}$ is also corrected for the readout correction process described in Bradley et al.[3].

## Note added

While finalizing this manuscript, two related preprints appeared that demonstrate destructive stabilizer measurements with a flag qubit[48] and flag fault-tolerant quantum error correction[49] with trapped-ion qubits. Furthermore, during the revision process, three related preprints appeared that demonstrate quantum error correction on a surface code using superconducting qubits[50,51] and realize a flag-based universal fault-tolerant gate set using trapped ions[52].

## Data availability

The underlying data and software code for generating the plots presented in the main text and Supplementary Information are available at Zenodo https://doi.org/10.5281/zenodo.6461872.

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

**Acknowledgements** This work was supported by the Netherlands Organisation for Scientific Research (NWO/OCW) through a Vidi grant and the Quantum Software Consortium programme (project no. 024.003.037/3368). This project has received funding from the European Research Council (ERC) under the European Union's Horizon 2020 research and innovation programme (grant agreement no. 852410). We gratefully acknowledge support from the joint research programme 'Modular quantum computers' by Fujitsu Limited and Delft University of Technology, co-funded by the Netherlands Enterprise Agency under project number PPS2007. This project (QIA) has received funding from the European Union's Horizon 2020 research and innovation programme under grant agreement no. 820445. This work is part of the project QCDA (with project number 680.91.033) of the research programme QuantERA, which is (partly) financed by the Dutch Research Council (NWO). This publication is part of the QuTech NWO funding 2020–2024 – Part I 'Fundamental Research' with project number 601.QT.001-1, which is financed by the NWO.

**Author contributions** M.H.A., Y.W. and T.H.T. devised the experiments. M.H.A. performed the experiments and collected the data. Y.W. and B.M.T. developed the fault-tolerant preparation scheme and its analysis. M.H.A., Y.W., J.R., B.M.T. and T.H.T. analysed the data. M.H.A., J.R., S.J.H.L. and C.E.B. prepared the experimental apparatus. M.M. and D.J.T. grew the diamond sample. M.H.A., Y.W. and T.H.T. wrote the manuscript, with input from all authors. B.M.T. and T.H.T. supervised the project.

**Competing interests** The authors declare no competing interests.

**Additional information**
**Correspondence and requests for materials** should be addressed to T. H. Taminiau.

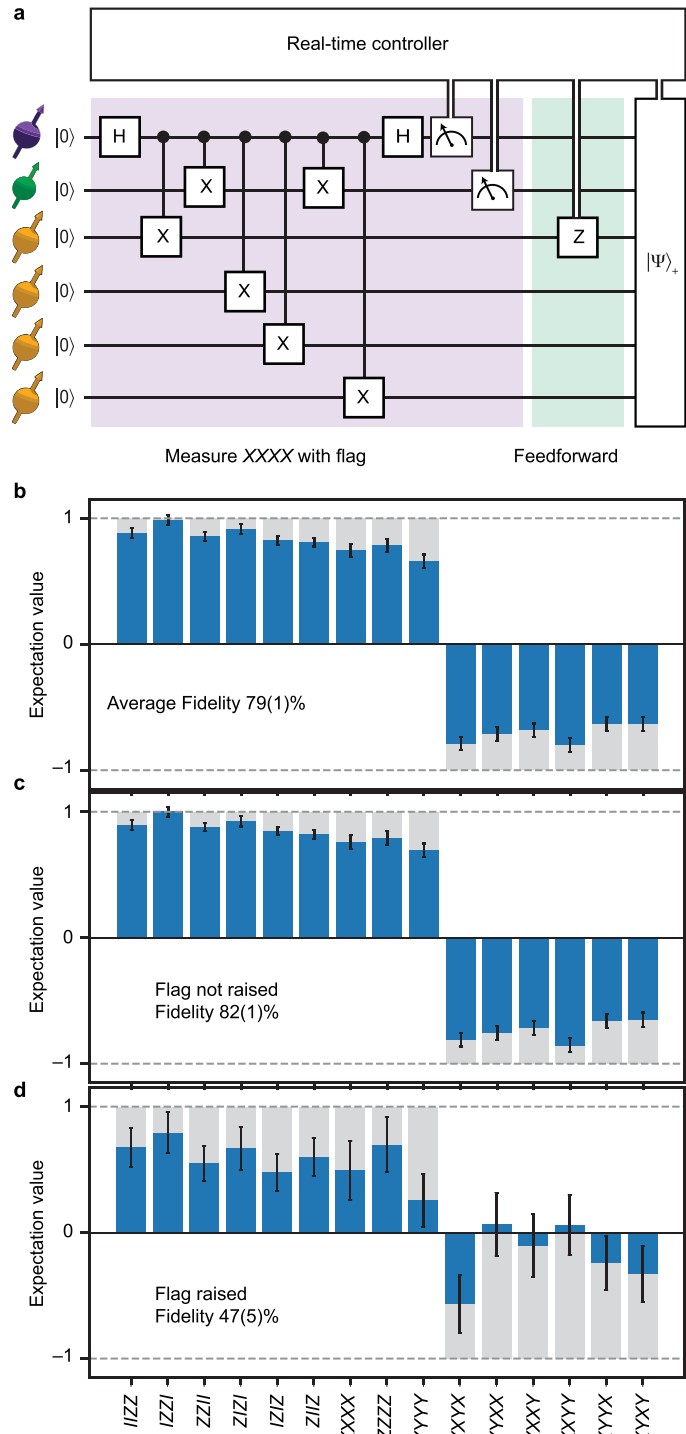

**Extended Data Fig. 1 | Non-destructive stabilizer measurements with a flag and real-time feedforward. a**, Circuit diagram for the deterministic preparation of a four-qubit GHZ entangled state $\left(|\psi\rangle_+ = (|0000\rangle + |1111\rangle)/\sqrt{2}\right)$ using a flagged measurement of the stabilizer *XXXX*. **b**, Measured expectation values of the 15 operators that define the ideal state. The average obtained fidelity is 0.79(1). **c**, Data post-selected on the flag not being raised. The obtained fidelity with the target state is 0.82(1). **d**, When the flag is raised, the obtained fidelity is 0.47(5). Grey bars show the ideal expectation values. Note that we perform this measurement as a test of the circuit, but that the flag information in this case does not carry any specific significance.

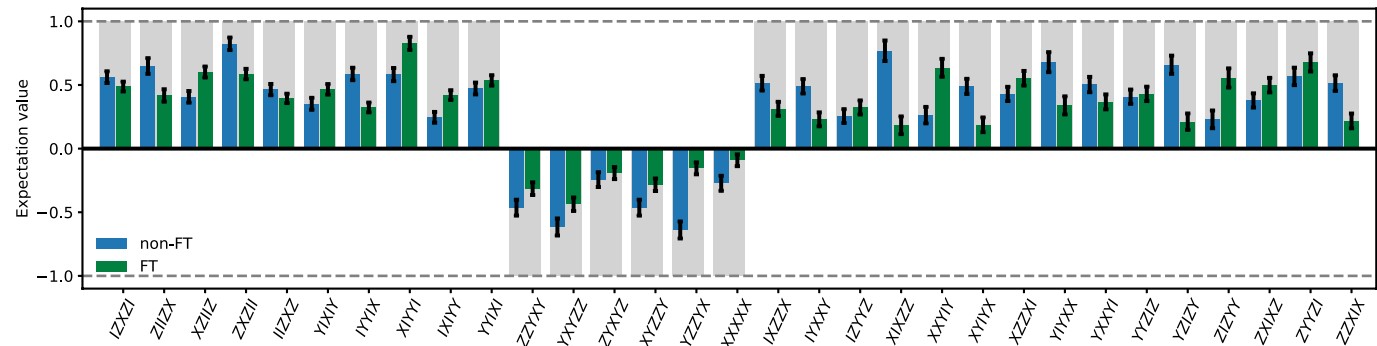

**Extended Data Fig. 2 | Measured expectation values for the encoded state.** Measured expectation values of the 31 operators that define the encoded state (for the circuit in Fig. 3). Grey bars show the ideal expectation values.