## [Peer Review File · Nature]

Manuscript Title: Fault-tolerant operation of a logical qubit in a diamond quantum processor

Reviewer Comments & Author Rebuttals

Reviewer Reports on the Initial Version:

Referees' comments:

Referee #1 (Remarks to the Author):

This paper describes the fault-tolerant operation of an NV centre coupled to 27 ^{13}C nuclear spins. This, in principle, provides a quantum computer of up to 29 qubits. To demonstrate fault-tolerant operations, the five-qubit quantum error correction code was implemented fault tolerantly on seven of these qubits, and single-qubit Clifford gate operations were demonstrated.

The results in this paper are novel, as the application of fault-tolerant methods to an NV-centre architecture, has not to my knowledge been demonstrated before. In addition, this paper demonstrates a couple of features that are currently under development for other architectures: One under-stated demonstration here is the ability to perform measurements and feed-forward corrections in real-time. This is a major milestone for any quantum computing architecture, and combining this with a genuinely fault-tolerant quantum error correction code seems to be an important achievement. Another is the adaption of flag qubits to the NV architecture. While these techniques are popular in the theoretical literature, this paper presents a robust adaption of these ideas to a physical system.

While this paper does indeed live up to its stated goals, it's also important to note that it does not demonstrate either quantum operations below the fault-tolerant threshold or a reduction of error below the physical error rate of the computer. Especially compared to reference 21 (which is now published in Nature, 598, p281–286, 2021 and could be updated in the bibliography) or reference 47, the error rates presented in this paper are much larger. This limitation is made clear enough in the conclusion, but in my humble opinion, it should be explicit about this point earlier in the paper (perhaps even in the abstract) as it could be misunderstood in this way. In our group, when this paper initially appeared on the arxiv, some of the researchers misread it in this way.

Similarly, it's obvious but important to note that single-qubit Clifford gates are clearly not universal for quantum computation and could be good to highlight what more is required to achieve universal fault-tolerant quantum computation. In particular, similar recent papers have demonstrated the fault-tolerant demonstration of magic states and two-qubit gates, and it would be good to be clear early in this paper that these are important aspects that will be required for fault-tolerant quantum computation not demonstrated here and are demonstrated on similar papers in different architectures.

A couple of very minor things: I believe what is referred to in the paper as multipartite entanglement, is often also called "genuine" multipartite entanglement (with a specific technical meaning), and I'm not totally sure about what it means to fall below 50% logical state fidelity in figure 5.

One query that I had, perhaps beyond the scope of this paper: I worry here that the large number of restarts required in state preparation exponentially badly with the size. The typical non-fault tolerant method to prepare such a graph state (or states locally equivalent to a graph state) is to

rotate the nuclear spins in an equal superposition and apply controlled-z gates between each consecutive pair. That method is not fault-tolerant, but could a similar preparation be combined with a measurement verification performed afterwards with much higher acceptance rates?

The paper is well written and clearly understandable. The data and methodology presented are appropriate for this paper. Similarly, the results are presented in an appropriate way with a thorough analysis and discussion of methods and errors.

What presented here makes good technical sense to me, as far as I know, is a novel result and appears well justified and sound. It suffers most from the authors (understandably) presenting their results in the best light, but not necessarily highlighting upfront some of the limitations of their results in comparison to similar papers appearing in the literature. I've tried to highlight these areas in my review. While I do think clearly acknowledging those limitations is important, I do not believe doing so should detract from the overall quality of this work.

Referee #2 (Remarks to the Author):

In this paper, the authors examine and implement the preparation, single qubit operation, and readout of a logical qubit in the $[[5,1,3]]$ code. This work leverages advances in fault tolerant circuits to enable this using only two auxiliary qubits, and only one qubit with preparation and measurement capabilities. This showcases both the tremendous progress made in the control and manipulation of individual spins in the proximity of a quantum sensor (in this case, the NV- defect in diamond), as well as the challenges in continuing towards greater successes. It also demonstrates for the first time a sufficiently large local register using NV centers with reasonably high fidelity to enable, e.g., distributed quantum computing protocols. Overall I find the work to be compelling and worthy of a broad audience.

What are the big questions in this type of work? First, the core operations that really matter for error correction include high quality two qubit gates, high quality state preparation, and rapid, high quality measurement that does not impact qubits not under measurement. The latter two elements are the most challenging for the NV-center system, given its single qubit (electron spin) that can be used for entropy reduction and for measurement. Given this natural bottleneck, and given challenges in high accuracy measurement, the question then becomes: how well can you do?

The authors approach this in stages. The first is to create a stabilizer state of a simple (albeit not very good) quantum code, the repetition code on four qubits. They can then show that their measurement system results can be fed-forward onto the C-13 qubits within the relevant, relatively long nuclear spin coherence times.

Their use of flag qubits is essential for getting closer to fault tolerance, and takes the system to its practical performance limits. This arises from the effects of optical measurement on the nearby N-14 nuclear spin, which leads to overall limits on the ability to jointly measure both at high fidelity.

Their basic conclusion, that this demonstrates the necessary components for fault tolerant preparation, manipulation, and readout of a single logical qubit, seems solid. That the audience may misinterpret this headline to mean much more than the present work needs to be recognized. While this is covered nicely in the conclusion, the authors should state clearly in the abstract that their logical qubit's performance is worse than the physical qubits, that is, that they are still well above the threshold for sustaining a logical qubit. This could be, e.g., the addition of a few words in the final sentence of the current abstract. I also do not believe that the inclusion of 'large-scale QIP' is warranted, given the demonstrated challenges and pitfalls for NV-NV coupling that have yet to be overcome.

Overall, I am a bit uncertain regarding the pitfalls they run into due to the echo sequence choices

necessary to refocus undesired couplings and the like. The pioneering NMR experiments of Vandersypen and Chuang did the community a favor by publishing the entire pulse sequence used, which highlighted the different refocusing sequences and the like. The inclusion of a similar diagram in the supplemental is a great service to the community, though this referee would love to see the whole circuit laid out in all its glory – taken together, there must be several thousand pulses used per successful experimental shot, including all the spin preparation steps in the C-13 nuclei, resets, etc.

Implicit in making this experiment work is the tremendous system calibration and characterization work done in Ref. 35. Also implicit is the control software and hardware that make it all possible. It would be helpful if they give references or pointers to the software and hardware used, and if open source, links where appropriate. For example, their compiler is not referenced nor described anywhere, yet is a key part of the experiment. How can I know whether it is working as intended? Are they intending to make any of the calibration systems and characterization systems available to the broader community?

I raise these points because this is, at its core, a systems paper. That is, the authors are using the work to demonstrate the successful operation of a complex control system with a quantum device at its core. For such complex systems, means of verifying their intended performance and validating the underlying elements are essential to ensure that the authors have not been misled by bugs or unintended effects. This is particularly challenging given that the only experimental data in the paper comprise fidelity estimates via tomographic reconstruction and estimates of expectation values of certain operators, which feed that reconstruction. Errors in interpretation or meaning from the underlying control system are thus uninterpretable to the expert.

This leads me to some concerns of fact, i.e., what the authors have actually done. For example, the tomography and fidelity statements made in the text seem imprecise at best. What method of density matrix reconstruction do the authors use to find their estimate of the density matrix given the tomographically complete set of measurements? Is there a MLE, some non-linear method, or a software toolkit they rely upon? What are error bars reported with respect to reconstructed values? How are they calculated?

For calculating fidelity, which norm do they infer? For example, Eq. (1) of the main text is not a typical fidelity measure, e.g., $\text{Tr}[(A^{1/2} B A^{1/2})^{1/2}]$ as would be expected when one quotes fidelity without further provisos. What do they mean by fidelity elsewhere in the text?

Given all the 'restart' parts of the different sequences, I would be curious to know:

(1) How many restarts are necessary, on average, to get to one successful state preparation. Would I be correct in multiplying the values in Table S1 to get this answer? I cannot tell from the caption.

(2) For prepared states, my understanding (based on Fig. S8) is that there are the following allowed states, depending upon whether a flag is raised, whether one of the flag-checked syndromes is raised, and whether a direct syndrome is raised (if one of the above two is raised). What are the relative probabilities of all these possible outcomes? The simplest way to address this would be to include the probabilities as observed of each branch in the flow chart. In the end I am curious about the yield overall, that is, the number of attempts to the number of successes.

Finally, given the large audience this paper will reach, I would encourage use of auxiliary qubit rather than ancilla qubit. The latter refers to female house slaves from Roman times, and may not be the look the authors are going for in their paper.

Author Rebuttals to Initial Comments:

Referees' comments:

Referee #1 (Remarks to the Author):

This paper describes the fault-tolerant operation of an NV centre coupled to 27 ^{13}C nuclear spins. This, in principle, provides a quantum computer of up to 29 qubits. To demonstrate fault-tolerant operations, the five-qubit quantum error correction code was implemented fault tolerantly on seven of these qubits, and single-qubit Clifford gate operations were demonstrated.

The results in this paper are novel, as the application of fault-tolerant methods to an NV-centre architecture has not to my knowledge been demonstrated before. In addition, this paper demonstrates a couple of features that are currently under development for other architectures: One under-stated demonstration here is the ability to perform measurements and feed-forward corrections in real-time. This is a major milestone for any quantum computing architecture, and combining this with a genuinely fault-tolerant quantum error correction code seems to be an important achievement. Another is the adaption of flag qubits to the NV architecture. While these techniques are popular in the theoretical literature, this paper presents a robust adaption of these ideas to a physical system.

We thank the referee for their time and for their detailed assessment of our work.

While this paper does indeed live up to its stated goals, it's also important to note that it does not demonstrate either quantum operations below the fault-tolerant threshold or a reduction of error below the physical error rate of the computer. Especially compared to reference 21 (which is now published in Nature, 598, p281–286, 2021 and could be updated in the bibliography) or reference 47, the error rates presented in this paper are much larger. This limitation is made clear enough in the conclusion, but in my humble opinion, it should be explicit about this point earlier in the paper (perhaps even in the abstract) as it could be misunderstood in this way. In our group, when this paper initially appeared on the arxiv, some of the researchers misread it in this way.

We thank the referee for pointing this out. To be clear upfront about the limitations of the results, and in order to avoid any potential initial misreading, we have now also incorporated the relevant statements from the conclusion paragraph in the abstract and introduction.

Additionally, we have removed the words “suppressing errors” in the second paragraph of the introduction, as it could be misinterpreted as pertaining to our results, rather than being general background information.

Finally, we have updated the bibliography for Egan et al (Nature, 598, p281–286, 2021) to its now published version.

Similarly, it's obvious but important to note that single-qubit Clifford gates are clearly not universal for quantum computation and could be good to highlight what more is required to achieve

universal fault-tolerant quantum computation. In particular, similar recent papers have demonstrated the fault-tolerant demonstration of magic states and two-qubit gates, and it would be good to be clear early in this paper that these are important aspects that will be required for fault-tolerant quantum computation not demonstrated here and are demonstrated on similar papers in different architectures.

The referee is right, of course, that single-qubit Clifford gates are not universal for quantum computation. While this is obvious for experts, we now added that the demonstrated gates are “non-universal” in the introduction and added a statement in the discussion of Fig.4 to clarify this for the broader audience targeted by this paper. In the same statement we now also explain that universal FT quantum gates would, for example, require an auxiliary logical qubit. We thank the referee for pointing out that this was not made explicit in the original submission.

While we are not aware of any previous work that experimentally realizes a universal fault-tolerant gate set, a recently appeared preprint using trapped ions has demonstrated this using flag qubits [Postler et al., arXiv:2111.12654v1, 24 Nov 2021]. We have added a note referring to this new work at the end of the manuscript.

A couple of very minor things: I believe what is referred to in the paper as multipartite entanglement, is often also called “genuine” multipartite entanglement (with a specific technical meaning),

We have changed ‘multipartite entanglement’ to ‘genuine multipartite entanglement’ in the caption of Figure 2.

and I'm not totally sure about what it means to fall below 50% logical state fidelity in figure 5.

A logical state fidelity below 50% in Fig. 5 simply means that the logical state has effectively flipped (after a perfect round of error correction). One is thus more likely to obtain the opposite logical state. Without a flag qubit, this is expected as the purposely inserted Y error on the ancilla qubit – on a system without any prior errors – causes a Y_3Y_5 error that propagates to a logical Z error.

One query that I had, perhaps beyond the scope of this paper: I worry here that the large number of restarts required in state preparation exponentially badly with the size. The typical non-fault tolerant method to prepare such a graph state (or states locally equivalent to a graph state) is to rotate the nuclear spins in an equal superposition and apply controlled-z gates between each consecutive pair. That method is not fault-tolerant, but could a similar preparation be combined with a measurement verification performed afterwards with much higher acceptance rates?

Indeed, FT preparation of logical states using the typical non-fault tolerant method to prepare graph states and then applying a conditional verification step is possible, as originally proposed by Chao and Reichardt in ref. 28. However, such a preparation scheme relies on multiple 2-qubit

gates between the data qubits (nuclear-nuclear spin gates), which is not optimal for our architecture, as nuclear-nuclear interactions are naturally weak. Instead, we introduce an alternative FT encoding scheme based on non-destructive parity measurements, which uses the native electron-nuclear spin gates. We have now added this information to Supplementary Information section VI.

For completeness, we also note that non-conditional FT state preparation can be constructed by following a non-fault tolerant state preparation circuit with several rounds of flag error corrections (Chamberland et al. ref. 29, as mentioned in Supplementary Information section VI), but the required number of parity measurements is considerable.

Second, the fact that the logical state preparation is heralded – a desired measurement outcome signals the successful preparation (although in a probabilistic way) – does not inherently limit scalability. This is analogous to heralded state preparation through entanglement distillation or magic-state distillation processes, which are ubiquitous elements in many large-scale quantum information schemes. The advantage of such a heralded scheme is that preparation fidelity can be increased at the cost of an overhead due to a lower success probability. How various schemes and trade-offs compare in hypothetical future larger-scale applications goes beyond the scope of this work. Likely, the optimal approach, and the desired trade-offs between overhead and fidelity, will generally depend on the particular application and system parameters.

The paper is well written and clearly understandable. The data and methodology presented are appropriate for this paper. Similarly, the results are presented in an appropriate way with a thorough analysis and discussion of methods and errors.

What presented here makes good technical sense to me, as far as I know, is a novel result and appears well justified and sound. It suffers most from the authors (understandably) presenting their results in the best light, but not necessarily highlighting upfront some of the limitations of their results in comparison to similar papers appearing in the literature. I've tried to highlight these areas in my review. While I do think clearly acknowledging those limitations is important, I do not believe doing so should detract from the overall quality of this work.

We thank the referee again for their detailed analysis of our work. We hope that the changes made – in particular the inclusion in the abstract and introduction of a summary of the limitations and future challenges that were originally discussed in the conclusion – have addressed their comments.

Referee #2 (Remarks to the Author):

In this paper, the authors examine and implement the preparation, single qubit operation, and readout of a logical qubit in the $[[5,1,3]]$ code. This work leverages advances in fault tolerant circuits to enable this using only two auxiliary qubits, and only one qubit with preparation and measurement capabilities. This showcases both the tremendous progress made in the control

and manipulation of individual spins in the proximity of a quantum sensor (in this case, the NV-defect in diamond), as well as the challenges in continuing towards greater successes. It also demonstrates for the first time a sufficiently large local register using NV centers with reasonably high fidelity to enable, e.g., distributed quantum computing protocols. Overall I find the work to be compelling and worthy of a broad audience.

We thank the referee for their time and their detailed review of our work.

What are the big questions in this type of work? First, the core operations that really matter for error correction include high quality two qubit gates, high quality state preparation, and rapid, high quality measurement that does not impact qubits not under measurement. The latter two elements are the most challenging for the NV-center system, given its single qubit (electron spin) that can be used for entropy reduction and for measurement. Given this natural bottleneck, and given challenges in high accuracy measurement, the question then becomes: how well can you do?

The authors approach this in stages. The first is to create a stabilizer state of a simple (albeit not very good) quantum code, the repetition code on four qubits. They can then show that their measurement system results can be fed-forward onto the C-13 qubits within the relevant, relatively long nuclear spin coherence times.

Their use of flag qubits is essential for getting closer to fault tolerance, and takes the system to its practical performance limits. This arises from the effects of optical measurement on the nearby N-14 nuclear spin, which leads to overall limits on the ability to jointly measure both at high fidelity.

For completeness and clarity, we note that we mitigate the effects of optical measurement on the ^{14}N nuclear spin qubit to a large extent by choosing it as the flag qubit. The flag qubit, unlike the other qubits, does not need to maintain coherence during the optical readout. We have added a sentence to clarify this in the manuscript.

Their basic conclusion, that this demonstrates the necessary components for fault tolerant preparation, manipulation, and readout of a single logical qubit, seems solid. That the audience may misinterpret this headline to mean much more than the present work needs to be recognized. While this is covered nicely in the conclusion, the authors should state clearly in the abstract that their logical qubit's performance is worse than the physical qubits, that is, that they are still well above the threshold for sustaining a logical qubit. This could be, e.g., the addition of a few words in the final sentence of the current abstract. I also do not believe that the inclusion of 'large-scale QIP' is warranted, given the demonstrated challenges and pitfalls for NV-NV coupling that have yet to be overcome.

We thank the referee for pointing this out. To be clear upfront about the limitations of the results, and in order to avoid any initial misinterpretation, we have now incorporated the relevant statements from the conclusion paragraph in the abstract and introduction as well.

We have also removed the word 'large-scale' from the abstract, as suggested. For completeness, let us mention that the potential scaling to large system sizes for these kind of systems (NV centers, and other optically active spins in wide-bandgap semiconductors) is envisioned to come from distributed quantum computation over optically connected networks, which can enable the required parallelized operation.

Overall, I am a bit uncertain regarding the pitfalls they run into due to the echo sequence choices necessary to refocus undesired couplings and the like. The pioneering NMR experiments of Vandersypen and Chuang did the community a favor by publishing the entire pulse sequence used, which highlighted the different refocusing sequences and the like. The inclusion similar diagram in the supplemental is a great service to the community, though this referee would love to see the whole circuit laid out in all its glory – taken together, there must be several thousand pulses used per successful experimental shot, including all the spin preparation steps in the C-13 nuclei, resets, etc.

We agree with the referee that an as complete as possible description of the pulse sequence is desirable. Unfortunately, the complexity of this experiment means that a complete visualization of the pulse sequence is not practical. The sequence includes 2000+ electron pulses, 50+ single-qubit RF pulses on the nuclear spins, 2000+ wait elements with precise nanosecond timings, as well as numerous optical pulses and logical triggers. Additionally, the real-time control and feed-forward means that actual sequence instances depend on the particular measurement outcomes. So, some level of abstraction into general instructions is required.

We have now carefully checked that all information and instructions to reconstruct the pulse sequences are given in the manuscript. For this, we have added pseudocode for the compiler and additional figures that clarify the timing of the pulse sequences, as well as the complete circuit diagrams. Additionally, we now provide the equations to precisely define all pulse shapes used (Methods).

Regarding pitfalls in the echo sequences, an important current limitation is that we decouple qubit number 3 from the other data qubits, but not the interaction between the other data qubits. More complex sequences that can decouple all nuclear-nuclear couplings are possible, but in the current system the available Rabi frequencies pose a limitation. Such improved decoupling sequences are anticipated to be feasible with future experimental upgrades to enable faster RF pulses (higher RF power using low-noise amplifiers, RF switches, and a better RF delivery to the sample (e.g. RF coils) to avoid sample heating).

We have now explicitly added this limitation and potential future solutions to the discussion of the echo sequences in Supplementary Section III G.

Implicit in making this experiment work is the tremendous system calibration and characterization work done in Ref. 35. Also implicit is the control software and hardware that make it all possible. It would be helpful if they give references or pointers to the software and hardware used, and if

open source, links where appropriate. For example, their compiler is not referenced nor described anywhere, yet is a key part of the experiment. How can I know whether it is working as intended? Are they intending to make any of the calibration systems and characterization systems available to the broader community?

I raise these points because this is, at its core, a systems paper. That is, the authors are using the work to demonstrate the successful operation of a complex control system with a quantum device at its core. For such complex systems, means of verifying their intended performance and validating the underlying elements are essential to ensure that the authors have not been misled by bugs or unintended effects. This is particularly challenging given that the only experimental data in the paper comprise fidelity estimates via tomographic reconstruction and estimates of expectation values of certain operators, which feed that reconstruction. Errors in interpretation or meaning from the underlying control system are thus uninterpretable to the expert.

We thank the referee for raising these detailed points and for pointing out that in this work the experimental system is an important result by itself. We have now extended the description of the experimental system and the compiler (details below). With these additions, we believe that we provide a complete description.

The most important part of the compiler is to translate the circuits (e.g. Supp Fig. 9) to the underlying pulse sequence, in particular the timing of the pulses. We have added detailed information on this compilation in the form of pseudocodes and additional figures to the supplementary materials. Note that one could go one level deeper in detail by including the codes to generate the actual hardware instructions, but those are very specific for the particular hardware (especially the waveform generators and microprocessors), and therefore of limited general use. The additions to describe the compiler are:

1. Supplementary Fig. 5: An example gate sequence illustrating the timing of the pulses and phase synchronization of the qubits (as used in the compiler).
2. Supplementary Fig. 9: A complete (example) circuit diagram for the experiment of Fig. 5, in the native gate set.
3. Supplementary section IX: Detailed pseudocode for the compilation of the circuit diagrams into the actual pulse sequences.

Additionally, we have added supplementary figures to provide more details about the experimental setup, and the calibration processes (summarized below):

1. Supp. Fig. 1: Sketch of the experimental setup including the control hardware.
2. Supplementary Fig. 7: The calibration of the systematic phase shifts that qubits acquire while applying an electron-nuclear two-qubit gate targeting other qubits.
3. Added panel d to Supplementary Fig. 4 to explicitly show how controlled-y gates (and other rotation axes) are constructed by using a controlled-x and phase-shift gates.

Regarding verification, the main validation comes from a comparison of the experimental results with the theoretical expectation. Note that many of the components and subroutines have been thoroughly tested in previous simpler experiments (see e.g. references 3 and 41). The novel capabilities in this work have been validated in intermediate experiments, such as the example shown in Fig. 2. While precise quantitative predictions for fidelities are outside of our current theoretical capabilities, our experimental observation of the theoretically expected behavior in complex experiments (see e.g. Fig. 3-5 and especially also Supplementary Figure 11) gives confidence that the system is operating as intended. We now explicitly provide the expected measurement outcomes for a system without decoherence in Supplementary Figure 11. Additionally, we now explain in the caption of Supplementary Fig. 8 that we use the QuTip Python toolbox for the simulations that are used to verify the compiled sequences and outcomes.

This leads me to some concerns of fact, i.e., what the authors have actually done. For example, the tomography and fidelity statements made in the text seem imprecise at best. What method of density matrix reconstruction do the authors use to find their estimate of the density matrix given the tomographically complete set of measurements? Is there a MLE, some non-linear method, or a software toolkit they rely upon? What are error bars reported with respect to in reconstructed values? How are they calculated?

For calculating fidelity, which norm do they infer? For example, Eq. (1) of the main text is not a typical fidelity measure, e.g., $\text{Tr}[(A^{1/2} B A^{1/2})^{1/2}]$ as would be expected when one quotes fidelity without further provisos. What do they mean by fidelity elsewhere in the text?

First, in this work we adopt the fidelity definition: $F(A, B) = \left(\text{Tr} \left(\sqrt{\sqrt{A} B \sqrt{A}} \right) \right)^+ = \text{Tr}(A, B)$, where A, B are the density matrices of the target and prepared states respectively. Note that this deviates by a square root from the distance metric given by the referee.

For the logical state fidelity, the definition was already explicitly given in the paper (Equation 1, which follows the above fidelity definition). For other state fidelities, like the GHZ state in Fig. 2, explicit definitions were missing. We have now added a section with the definition of the state fidelity used to the Methods.

Second, note that we do not perform state tomography. For stabilizer states (such as the ones prepared in this paper) one does not need state tomography to obtain the desired fidelity measures. Instead, it is sufficient to measure the expectation values of the 2^{n-1} non-zero operators that define the prepared state (with n the number of qubits). These operators can be generated from the group of stabilizers defining the state [see e.g. ref. 20]. Therefore, we do not aim to reconstruct the density matrix and no MLE or special software toolkits are required for our analysis.

The explicit equations to obtain the fidelities are given in equations 4,5 and 6. For completeness, we now also provide the error distribution in the prepared state (i.e., Fig. 3b,c, equations 10 and

11) written out explicitly in terms of the measured expectation values (supplementary equations S9 - S12).

In the original manuscript, we had used the word “tomographic” to distinguish the final measurements to characterize the output state from the non-destructive parity measurements that detect errors. We now see that this can be confusing and have removed all instances of the word “tomographic”.

The error bars in the reported fidelities and probabilities are calculated using direct error propagation from the errors in the measured expectation values, which originate from a binomial process. We have now added a note about error bar calculations to Methods to clarify this.

Given all the ‘restart’ parts of the different sequences, I would be curious to know: (1) How many restarts are necessary, on average, to get to one successful state preparation. Would I be correct in multiplying the values in Table S1 to get this answer? I cannot tell from the caption.

Indeed, the average number of restarts to get one successful state preparation is obtained by multiplying the values in Table S1. We have now added the total success probability and adapted the table caption to clarify this.

For completeness, we note that the success probability is strongly impacted by our choice of running the non-FT part of the encoding circuit (i.e. measuring p_3 , p_4 , p_5) in a conditional way. This part of the circuit can be performed by continuing on both measurement outcomes (using the feedforward capability demonstrated in our work). However, as explained in the Methods section “Readout of the auxiliary qubit”, we chose to accept only the +1 outcomes (NV electron spin in $|0\rangle$), because these are more reliable. This yields a slightly higher fidelity, but comes with the cost of a lower success probability.

(2) For prepared states, my understanding (based on Fig. S8) is that there are the following allowed states, depending upon whether a flag is raised, whether one of the flag-checked syndromes is raised, and whether a direct syndrome is raised (if one of the above two is raised). What are the relative probabilities of all these possible outcomes? The simplest way to address this would be to include the probabilities as observed of each branch in the flow chart. In the end I am curious about the yield overall, that is, the number of attempts to the number of successes.

All the relevant probabilities in our experiments (for FT state encoding and FT stabilizer measurement outcomes) are given in Fig. 5b and Supplementary Table 1. The total success probability is now explicitly given in Supplementary Table 1.

Note that Fig. S8 (Fig. S12 in the revision) represents the complete QEC protocol (after encoding) as proposed by ref. 28. We do not experimentally characterize this entire demanding process. Instead, we show this figure in the supplementary information for pedagogical reasons as it outlines how our experimentally demonstrated flagged stabilizer measurements provide a primitive for FT quantum error correction and how the error syndrome is re-interpreted upon

different flag measurement outcomes (as described in the methods section “Logical state fidelity with flag”).

Finally, given the large audience this paper will reach, I would encourage use of auxiliary qubit rather than ancilla qubit. The latter refers to female house slaves from Roman times, and may not be the look the authors are going for in their paper.

We have replaced the word ancilla by auxiliary throughout the paper.

List of other changes made upon revision:

- 1) We have added a note with two related preprints on error correction with the surface code using superconducting qubits that recently appeared: Krinner et al., arXiv:2112.03708, 7 Dec 2021, and Zhao et al, arxiv:2112.13505, 27 Dec 2021.
- 2) For completeness, in the introduction, we have added a reference to a relevant paper that was left out in the first version: Erhard et al., Nature 589.7841 (2021): 220-224.

Reviewer Reports on the First Revision:

Referees' comments:

Referee #1 (Remarks to the Author):

Thank you for your changes and replies, which address the queries in my previous review very well.